# Annual Crops Contribute More Predators than Perennial Habitats during an Aphid Outbreak

**DOI:** 10.3390/insects14070624

**Published:** 2023-07-11

**Authors:** Crystal D. Almdal, Alejandro C. Costamagna

**Affiliations:** Department of Entomology, University of Manitoba, 217 Animal Science/Entomology Building, 12 Dafoe Road, Winnipeg, MB R3T 2N2, Canada; cdalmdal@gmail.com

**Keywords:** adjacent habitat, soybean aphid, predator movement, Syrphidae, Coccinellidae

## Abstract

**Simple Summary:**

Understanding how insect predators utilize different crops and natural habitats is crucial to improve the ecological service of pest control in agricultural landscapes. The aim of this study was to develop a better understanding of how adjacent habitats contribute predators during pest outbreaks. We used a soybean aphid outbreak to test how adjacent habitats impact predator movement and abundance in soybean. We found that hoverflies and ladybeetles were the most common predators moving into soybean and their movement was related to the presence of aphids. In general, we found annual crops had more predators than a perennial crop or habitat. Our study suggests that adjacent wheat and canola fields result in more predators moving into soybean to control aphids than adjacent alfalfa or woody vegetation.

**Abstract:**

Crops and semi-natural habitats provide predator populations with varying floral and prey resources, but their individual role on predator movement has seldom been studied. Here, we tease apart the role of adjacent habitats, predator abundance in the adjacent habitat, and soybean aphid (*Aphis glycines* Matsumura) abundance in soybean (*Glycine max* (L.) Merr.) on predator movement into soybean. We studied 12 soybean fields adjacent to alfalfa (*Medicago sativa* L.), canola (*Brassica napus* L.), spring wheat (*Triticum aestivum* L.), or woody vegetation, during a soybean aphid outbreak. Bidirectional Malaise traps and sticky traps were used to quantify predator movement between and abundance within soybean and adjacent habitats, respectively. Field plant counts were conducted to quantify aphid abundance in soybean. Coccinellidae and Syrphidae were the two most abundant families collected. Coccinellids and *Eupeodes americanus* (Wiedemann) (Diptera: Syrphidae) had net movement in soybean and their movement increased with aphid abundance. Movement of *E. americanus* was highest from wheat, coccinellid abundance was higher in wheat than woody vegetation, *Toxomerus marginatus* (Say) (Diptera: Syrphidae) abundance was highest in canola, and all other predators were more abundant in canola than woody vegetation. In general, our study suggests that annual crops have and provide more predators to soybean during aphid outbreaks than perennial habitats.

## 1. Introduction

Predators have two main resource requirements to survive and reproduce: food and shelter. Food resources include crop pests and alternative prey and non-prey items such as nectar and pollen, and shelter resources include sites that provide refuge during times of disturbance and for overwintering [1,2]. Generalist predator assemblages have varying resource requirements and may require multiple resources throughout their life cycle [3,4]. Insects that are predaceous as larvae and anthophilous as adults will have different resource requirements than those who are predaceous as both larvae and adults, requiring several habitats to complete their life cycle [4]. Therefore, ease of movement between habitats may be crucial for obtaining the resources necessary for development. 

Habitats may be utilized by generalist predators at different times of the growing season. Perennial habitats such as semi-natural habitat or alfalfa may be important early season habitats for predators as they provide floral and prey resources before flowering and pest establishment in managed fields [5]. Additionally, perennial habitats are thought to provide shelter during the growing season from pesticide applications and harvesting activities and at the late phase of the growing season for overwintering [2,6,7]. Alternatively, annual crops may provide floral and prey resources in the mid and late portion of the growing season [5]. Therefore, adjacent habitats may be important in providing generalist predators with food and shelter resources during times of low prey availability and disturbance events and be important contributors of predators to control pest populations [6,7,8].

Thus far, most studies on the movement of aerial natural enemies have focused on movement between managed land and adjacent semi-natural habitats [8,9,10,11,12]. Natural enemies have been observed moving from semi-natural habitat to adjacent crops early in the season [13], and from crops to semi-natural habitat late in the season [8,9,10,11,12] and during times of disturbance [7]. However, movement of natural enemies between other perennial habitats and annual crops in the landscape may also be important, especially in landscapes dominated by crops [11,13,14,15], but has received less attention [13,16]. Indirect evidence of the role of different crops comes from studies showing that crop diversity in the surrounding landscape benefits natural enemy populations [17,18,19], and is associated with lower pest abundances [20,21] and increased aphid suppression [22]. However, direct evidence on the function of different habitats and crops in contributing predators to adjacent crops, particularly during pest outbreaks, is lacking.

In this study, we compared aphidophagous predator movement between soybean (*Glycine max* (L.) Merr.) and four adjacent habitats during an outbreak of the soybean aphid (*Aphis glycines* Matsumura), which is a rare event in Manitoba [22,23]. We selected adjacent habitats that may provide different resources for aphidophagous predators: two annual crops, canola (*Brassica napus* L.) and spring wheat (*Triticum aestivum* L.), a perennial crop, alfalfa (*Medicago sativa* L.), and a perennial habitat, woody vegetation. Specifically, we asked: (1) Do adjacent habitats affect predator movement into and out of soybean? (2) Does the role of adjacent habitats differ for predators that have anthophilous adults? and (3) Do other factors affect the movement of predators into soybean? (i.e., specifically, (a) predator abundance in adjacent habitats, and (b) soybean aphid abundance). We expected canola would mostly provide pollen and nectar resources [24], and benefit more anthophilous predators (syrphids, chrysopids); wheat would provide aphids [25] and benefit more non-anthophilous predators (coccinellids, hemerobiids, anthocorids, nabids, etc.); alfalfa would provide aphids, pollen and nectar [3,26] benefiting all aphidophagous predators, and woody vegetation would provide shelter and early season resources, but its benefit would be limited for predators at the time of our study [13,27]. In addition, we expected predators that are aphidophagous throughout their life would have net movement into soybean and be mostly influenced by aphid abundance compared to anthophilous syrphid adults, who we expect would be mostly influenced by the adjacent habitat type and their abundance in the adjacent habitat. Our study occurred when soybean was vulnerable to aphid damage and population numbers were approaching or surpassing economic thresholds [28]. This study provides insight on how adjacent habitats contribute predators during unique outbreak events.

## 2. Materials and Methods

### 2.1. Site Selection

Twelve soybean field sites were selected in 2017 in Manitoba, Canada. Soybean fields were adjacent to either canola, spring wheat, alfalfa, or woody vegetation (*n* = 3 for each adjacent habitat type; Appendix A). During the time of sampling, soybean was between full bloom and beginning pod stage (Appendix A), canola was in the late flowering and seed development stage, wheat was ripening, and alfalfa was in the beginning to mid-flowering stage. Woody vegetation included saskatoon (*Amelanchier alnifolia* (Nutt.) Nutt.) and chokecherry (*Prunus virginiana* L.) bushes, a grassy understorey, and the overstorey was dominated by deciduous trees, mainly white poplar (*Populus* spp.), oak (*Quercus* spp.), Manitoba maple (*Acer negundo* L.), and American elm (*Ulmus americana* L.).

### 2.2. Insect Sampling

Double sided yellow sticky traps (18 × 14 cm; Alpha Scent, Inc., West Linn, OR, USA) were used to sample predator abundance in each habitat and bidirectional Malaise traps (Townes style, 190 cm front height × 160 cm length × 110 cm back height; Sante Traps, Lexington, KY, USA) were used to sample predator movement between habitats. Bidirectional Malaise traps have been successfully used to track the movement of insects, including bees [29], parasitoids [10], and various agricultural pests and their natural enemies [13,14,16,22,23,30]. Traps were set up the third week in July for a period of two weeks (removed first week of August), at the time soybean aphid populations were abundant and likely to cause economic damage [31]], and insects were sampled weekly (*n* = 2). Adults of the following predator families were identified: Anthocoridae, Chrysopidae, Coccinellidae, Hemerobiidae, Nabidae, Staphylinidae, and Syrphidae as these groups have been shown to attack and provide strong control of soybean aphids [32,33,34,35,36]. Five sticky traps were placed 5 m apart, in soybean 15 m from the adjacent field border (*n* = 5) and in the adjacent habitat 20 m from the soybean field border (*n* = 5) parallel with the shared border. Sticky traps were set at the height of the surrounding vegetation using bamboo stakes and twist ties. Upon collection, sticky traps were wrapped in plastic and preserved in freezer conditions (−18 °C) prior to and after identification (*n* = 240). Predator counts were summed within each site and habitat type (*n* = 48; Appendix A). One bidirectional Malaise trap was placed in the field border between soybean and its adjacent habitat, centered around sticky trap placement, to quantify the movement of aphid predators into and out of soybean (*n* = 48). In each field, the Malaise trap was placed approximately 75 m from the road border. The following references were used to identify specimens from the Malaise and sticky trap samples: coccinellids [37,38] and chrysopids [39,40]. Hemerobiids were identified to species in Malaise traps but only to family in sticky traps due to difficulties distinguishing wing venation on the traps [41,42]. For syrphids, *Toxomerus marginatus* (Say) was identified in both Malaise and sticky trap samples, and *Eupeodes americanus* (Wiedemann) was identified to species in Malaise traps [43] but identification by genitalia was not possible in sticky trap samples. We conducted destructive plant counts weekly for two weeks in soybean to determine aphid abundance (*Aphis glycines* M.), in an area of approximately 75 m × 75 m, near the sticky traps. A haphazard process was used to select plants, in which a flag was thrown in the soybean field and the nearest plant to the flag was chosen. Twenty plants were selected per field, unless plants had over 1000 aphids/plant on average, then only 10 plants were selected [31].

Due to logistical reasons, two field sites were sampled after six days, and one site was sampled after eight days instead of seven during the second week of our study. Therefore, predator captures from bidirectional Malaise traps and sticky traps were adjusted to seven days in these sites by dividing predator abundance by the number of days sampled and multiplying by 7. In addition, aphid field plant counts were adjusted to seven days to correspond with the adjustments made for predator abundance. This was carried out by calculating the intrinsic rate of growth from the predator exclusion cages used in a separate experiment [22]. Briefly, each soybean field had five predator exclusion cages placed 20 m from the adjacent habitat border to measure aphid growth unimpacted by predator populations. Cages ran parallel to the adjacent habitat and were separated by 10 m. Each cage consisted of a pot with two soybean plants surrounded by a wire tomato cage (0.4 m width × 1 m height) and fine mesh (0.24 mm^2^ no-see-um netting) that prevented aphid and predator movement. Each cage was infested with 14 sentinel soybean aphids the third week of July and aphid counts were conducted weekly for two weeks [22]. Aphid counts were adjusted assuming exponential growth, using Equation (1),
*r* = *ln*(*N_t_*) − *ln*(*N*_0_)/*t*(1)
which includes the population size at exclusion cage set up (*N*_0_) and removal (*N_t_*), and the number of days since set up (*t*) [22]. Aphid abundance in soybean at 14 days was then adjusted using the intrinsic rate of growth calculated from the exclusion cage using Equation (2)
*N*_14days_ = *N*_7days_*^rt^*(2)

### 2.3. Statistical Analysis

All statistical analyses were performed in R [44]. We divided our statistical analyses into three steps to avoid overparameterization of models. First, linear models were constructed to test whether predator abundance in soybean differed as a function of adjacent habitat type (alfalfa, canola, woody vegetation, or spring wheat). Similar models were constructed to test whether predator abundance in the adjacent habitat differed as a function of habitat type. Models only included data from the second sampling week, as the first week had very low captures at most sites (*n* = 12). Response variables in both models included the number of syrphids, *T. marginatus*, coccinellids, and all other predators combined for predators in soybean and in the adjacent habitat from sticky traps. All other predators combined included summed predator counts of members from the families Anthocoridae, Nabidae, Chrysopidae, Hemerobiidae, and Staphylinidae, which had limited numbers of individuals captured across both sticky traps and Malaise trap samples. The Box–Cox method was performed to select an appropriate transformation for each model to meet model assumptions of normality [45]. A one-way ANOVA with field as a blocking factor was conducted to determine if the mean number of aphids/plants differed between the two sampling weeks. 

Second, factors affecting predator bidirectional movement were investigated to determine whether predators had higher movement into than movement out of soybean and how adjacent habitats impact overall predator movement in soybean. We tested this by constructing three separate linear mixed effects models by sequentially adding explanatory variables: (1) movement direction, (2) movement direction and adjacent habitat type, and (3) the two previous factors and their interaction. Week was included as a fixed covariate as it only had two levels and the field site was a random factor in all models to account for the two directions of the trap and the two sampling weeks. Model selection using Akaike information criteria (AIC) was performed to select the best model fit with the maximum likelihood estimation. The AIC was corrected for small sample sizes using the package ‘*AICcmodavg*’ [46]. Models were selected if there was a significant improvement in model fit based on *p*-values from likelihood ratio tests using the *anova* function in R and an alpha level of 0.05. Selected models were then fit again using restricted maximum likelihood to achieve unbiased estimates of variance [47]. Response variables included bidirectional movement (individuals/trap) of syrphids, *T. marginatus*, *E. americanus*, coccinellids, and all other predators combined. 

Finally, we fit linear mixed effects models to test how predator movement into soybean (i.e., unidirectional movement) was affected by adjacent habitat type, while controlling for predator abundance in the adjacent habitat (predators/5 sticky traps) and aphid abundance in soybean (mean # aphids/plant). *Eupeodes americanus* was not identified in sticky trap samples, therefore, only field aphids and adjacent habitat type were included as predictors. Field site was included as a random factor in all models to account for the two sampling weeks. Response variables were the same as in the previous analysis but only included individuals captured moving into soybean (individuals/trap). 

All linear mixed effects models were constructed in the ‘nlme’ package [48], the ‘emmeans’ package was used for pairwise comparisons of adjacent habitat types using a *post hoc* Tukey test [49]. Model residuals were plotted to visually check model assumptions of normality and homogeneity of variances. All predator variables were (log10) + 1 transformed (syrphids, coccinellids, all other predators combined, *T. marginatus*, and *E. americanus*), and aphid abundances were log10 transformed to meet model assumptions. Field means ± standard errors are presented.

## 3. Results

Over the two sampling weeks, we observed on average 551.04 ± 145.09 aphids/plant in soybean from field plant counts, and aphid abundance was higher the second sampling week (833.25 ± 256.65) than the first (268.83 ± 87.61; F_1,11_ = 39.18, *p* < 0.0001). Syrphids were the most abundant predator group in soybean with over 11,000 syrphids captured, followed by coccinellids with over 400 individuals captured on sticky traps over the two sampling weeks (Table 1). *Harmonia axyridis* (Pallas) was the most abundant coccinellid, followed by *Coccinella septempunctata* L. and *Hippodamia tredecimpunctata* (L.) (Appendix A). Fewer anthocorids, staphylinids, and chrysopids were captured, and no nabids were collected in soybean (Appendix A). Over 31,000 syrphids were collected from bidirectional Malaise traps over the two weeks (Appendix A). *Eupeodes americanus* was the most abundant syrphid (60%) followed by *T. marginatus* (33%). We collected 458 staphylinids and 459 coccinellids, with *Coccinella septempunctata* (68%), *H. tredecimpunctata* (13%), and *H. axyridis* (10%) the most abundant coccinellids found moving into and out of soybean. Intermediate numbers of chrysopids (134) and hemerobiids (68) and low numbers of nabids (26) and anthocorids (21) were captured.

In soybean, the abundance of syrphids (cube root transformed; F_3,8_ = 1.96, *p* = 0.20), *T. marginatus* (log transformed; F_3,8_ = 1.19, *p* = 0.37), coccinellids (square root transformed; F_3,8_ = 0.37, *p* = 0.78), and all other predators combined (log transformed; F_3,8_ = 0.12, *p* = 0.95) did not differ as a function of adjacent habitat type (Table 1). Adjacent habitats significantly affected the abundance of all predator groups (Table 1): syrphids (cube root transformed; F_3,8_ = 6.00, *p* = 0.019) were higher in canola than alfalfa (*p* = 0.017) and woody vegetation (*p* = 0.045); *T. marginatus* (cube root transformed; F_3,8_ = 16.90, *p* = 0.0008) was higher in canola than in alfalfa (*p* = 0.019), wheat (*p* = 0.0068) or woody vegetation (*p* = 0.0009); coccinellids (square root transformed; F_3,8_ = 4.85, *p* = 0.033) were higher in wheat than woody vegetation (*p* = 0.044); and all other predators combined (log10 + 1 transformed; F_3,8_ = 4.34, *p* = 0.043) were almost significantly higher in canola (*p* = 0.064) and wheat (*p* = 0.052) than in woody vegetation.

No significant interactions between adjacent habitat and movement direction were observed among all tested models (Appendix A). The best bidirectional movement model for syrphids, *E. americanus*, and coccinellids was the model with movement direction, week, and adjacent habitat, and for *T. marginatus* and all other predators combined was the model with movement direction and week (Appendix A). All predator groups tested had higher movement the second sampling week except for all predators combined (Table 2). We found syrphids, *E. americanus*, and coccinellids had higher movement into soybean than out of soybean, and *T. marginatus* and all other predators combined were unaffected by movement direction (Table 2). Bidirectional movement of syrphids and *E. americanus* was affected by adjacent habitat type (Table 2), with movement lower in woody vegetation compared to wheat (Syrphid *p* = 0.043, Figure 1a; *E. americanus p* = 0.035, Figure 1b), and *E. americanus* movement lower in alfalfa (*p* = 0.049; Figure 1b) compared to wheat. By contrast, movement of *T. marginatus* was unaffected by adjacent habitat type (Table 2 and Appendix A). Coccinellid bidirectional movement was marginally impacted by adjacent habitat (Figure 1c) and movement of all other predator families combined was similar across both movement directions (Table 2).

Syrphid movement into soybean was positively associated with syrphid abundance in the adjacent habitat (Figure 2a), aphid abundance in soybean had no effect, and movement into soybean differed among adjacent habitats (Table 3); however, there was not enough power to detect these differences using pairwise comparisons (Figure 2b). Movement of *E. americanus* into soybean was affected by both the abundance of aphids in soybean (Figure 2c) and by the adjacent habitat (Table 3), with alfalfa (*p* = 0.042) and woody vegetation (*p* = 0.023) having fewer individuals moving into soybean than wheat (Figure 2d). Movement of *T. marginatus* into soybean was only affected by the number of *T. marginatus* in the adjacent habitat (Figure 2e). Coccinellid movement into soybean was positively affected by aphid abundance (Figure 2f); however, adjacent habitat type and the number of coccinellids in the adjacent habitat had no effect (Table 3). Movement of all other predator families combined into soybean was unaffected by adjacent habitat, their abundance in the adjacent habitat, and aphid abundance in soybean (Table 3). 

## 4. Discussion 

Overall, we found that aphidophagous predators moving between soybean and adjacent habitats were dominated by syrphids, followed by coccinellids and then fewer individuals of other groups. We observed that adjacent habitat type affected the overall movement of syrphids, but only marginally affected coccinellids. In general, movement into soybean was higher than movement out of soybean, but the factors affecting movement into soybean varied among the main predator groups. This included positive effects of the number of syrphids in the adjacent habitat and wheat on syrphid movement into soybean, and aphid abundance in soybean on syrphid and coccinellid movement into soybean. Therefore, we demonstrate anthophilous syrphid adults were more affected by adjacent habitat type than predacious coccinellid adults, but contrary to our predictions, some species of anthophilous predators were also moving to soybean in response to aphid abundance. In general, we found that annual crops contributed to higher predator abundance and movement than perennial habitats. 

Coccinellid movement into soybean was driven by aphid abundance. Coccinellids had higher movement into soybean than out of soybean, and higher levels of movement were observed during the second week of our study, coinciding with higher aphid abundances. Previous studies have shown that coccinellid movement is positively associated with aphid abundance [50], that they aggregate in areas with high aphid densities [15,51,52,53], and have higher movement into soybean than out of soybean even in low aphid years [16]. We expected wheat and alfalfa to contribute more coccinellids to soybean than canola and woody vegetation as we anticipated these habitats would provide aphids early in the season before aphid establishment in soybean [16,25,26]. Although coccinellids were more abundant in wheat than woody vegetation, our study showed little evidence of effects of adjacent habitat type or coccinellid abundance in the adjacent habitat on coccinellid movement into soybean, suggesting coccinellids may be moving from further distances to reach soybean infested fields. This is supported by previous results from mark–release–recapture experiments that found coccinellids travelling long distances for food when prey availability was low [54]. In summary, our study provides direct evidence that coccinellids are highly responsive to aphid density and demonstrates that their positive response occurs regardless of the adjacent habitat type or their abundance in adjacent habitats, suggesting they locate aphids at larger spatial scales.

In our study, the coccinellid community was primarily composed of two exotic and one native species. Although previous studies have demonstrated exotic coccinellids can disrupt and displace some native coccinellid species in agricultural landscapes [55,56,57], we show that the native coccinellid *H. tredecimpunctata* was abundant and responded in high numbers to aphid density in soybean alongside exotic *H. axyridis* and *C. septempunctata*. *Hippodamia tredecimpunctata* has been shown to be similarly unaffected by exotic coccinellids in Michigan; however, this species was rarely observed [55,57]. In Manitoba, *H. tredecimpunctata* was the second most abundant coccinellid moving in soybean when aphid abundance was low [16]. Therefore, we provide further evidence that *H. tredecimpunctata* remains unaffected by exotic coccinellids and can respond to aphid populations in soybeans in Manitoba. 

The two most common syrphids had contrasting responses to soybean aphid density and were differentially impacted by the type of adjacent habitat. Moreover, their varied response to aphid density is likely why we see higher movement of *E. americanus* into than out of soybean but no directionality in the movement of *T. marginatus*. In our study, *E. americanus* and *T. marginatus* comprised 92% of the syrphids collected in Malaise traps, with *E. americanus* 1.8-fold more abundant than *T. marginatus* (Appendix A). Both species are anthophilous as adults, have aphidophagous larvae that feed on *Aphis* species, and have been observed attacking aphids in soybean [58,59,60]. When soybean aphid abundance is low, *T. marginatus* makes up 90% and *E. americanus* less than 3% of syrphid movement in soybean, in Manitoba [16]. Here, we show *E. americanus* increases its bidirectional movement in soybean and becomes more abundant than *T. marginatus* in an outbreak year, suggesting *E. americanus* is more responsive to aphid outbreaks than *T. marginatus.*

Movement of *E. americanus* into soybean was in response to both aphid abundance and adjacent habitat type. This species is common in North America, feeds on a variety of plant species (60 reported), and its larvae are generalist aphid predators [59,61]. *Eupeodes americanus* has been shown to respond numerically to woolly apple aphid, *Eriosoma lanigerum* (Haussman) [62] and balsam twig aphid, *Mindarus abietinus* Koch (Hemiptera: Aphididae) abundance [63]. Here, bidirectional movement of *E. americanus* was higher the second sampling week, coinciding with higher aphid abundance and changes in plant phenology in canola and wheat. Aphid honeydew has been shown to be an important oviposition cue for syrphids and is often substituted for nectar when it is scarce [64,65,66]. Therefore, *E. americanus* may have been moving to soybean to oviposit, mate and/or to feed on aphid honeydew since wheat was ripening and canola was transitioning from the flowering to pod stage at the time of sampling.

Canola was one of the main crops contributing to *E. americanus* movement in soybean, which we expected, as it provides a vast area of floral resources [5,67,68,69]. Canola was in the late flowering stage/early pod stage and syrphids have been observed to increase movement from canola in response to changes in canola phenology [67], therefore *E. americanus* was likely moving to soybean for floral and prey resources. Wheat was also a major contributor of *E. americanus* and adults were likely moving to soybean to oviposit in response to aphid abundance [67,68,69]. We hypothesize that wheat may have harbored aphids earlier in the season and been an oviposition site, since it was in the ripening stage and flowers were no longer present [68,70,71]. The lower movement of *E. americanus* from alfalfa than wheat may be due to *E. americanus* feeding on alfalfa, as previous studies have observed *E. americanus* feeding on species of *Medicago* [72]. We expected few syrphids to move between woody vegetation and soybean since syrphids mainly use semi-natural habitat for feeding and oviposition sites during times of low floral and prey resources in crop fields [67,69]. Although we were unable to test the contribution of the abundance of *E. americanus* in the adjacent habitat on its movement into soybean, the impact of syrphid abundance in the adjacent habitat on overall syrphid movement into soybean suggests there would be a positive association, since syrphid movement into soybean is primarily composed of *E. americanus* individuals (Appendix A). In addition, syrphids were more abundant in canola than alfalfa and woody vegetation, which is likely why we see higher movement of *E. americanus* in soybean associated with canola and wheat. In summary, our results suggest canola and wheat are the main contributors of *E. americanus* to soybean during aphid outbreaks, providing complementary resources to this abundant syrphid species.

Movement of *T. marginatus* was similar in both directions and the type of adjacent habitat had no impact on their movement. Similarly, movement of *T. marginatus* into soybean was unaffected by aphid abundance and adjacent habitat type even though they were more abundant in canola than alfalfa, wheat, and woody vegetation, and instead their movement was associated to the number of *T. marginatus* in the adjacent habitat. This species is abundant and widespread in North America, and adults feed on a variety of plants (114 reported), but larvae feed on relatively few aphid species [16,58,61]. When *T. marginatus* was more abundant in the adjacent habitat, we saw passive diffusion of *T. marginatus* in soybean. In general, syrphids tend to oviposit in fields with high aphid densities [58,70]. Therefore, their lack of response to aphid density was surprising, since their bidirectional movement was higher the second sampling week and their abundance was higher when there were more aphids in soybean [16,22]. Our results show that *T. marginatus* moves into soybean in high numbers when its abundant in the adjacent habitat, and moves interchangeably in and out of soybean, suggesting *T. marginatus* may move to soybean to oviposit and/or mate and move to other habitats to feed on nectar resources.

All other predator groups combined had similar movement in both directions, and their movement into soybean was not affected by aphid abundance or adjacent habitat type, although their abundance was higher in canola and wheat than woody vegetation. Samaranayake and Costamagna [16] also found movement of anthocorids, hemerobiids, and nabids was unaffected by adjacent habitat type and that they moved similarly into and out of soybean. Combining predator families into one grouping was carried out due to low capture rates; however, not all families included had the same feeding and oviposition strategies, and thus may be differentially affected by habitat function, which has been shown to affect predator movement [15,73,74,75]. However, we note that the densities of these other predator groups were 1.4-fold (Appendix A) greater during this outbreak year than in previous low aphid years [23,76], suggesting that the lack of response to adjacent habitats found in our study is not an artifact of low predator density. In summary, we observed that predators other than syrphids and coccinellids had low capture rates, moved interchangeably between soybean and its adjacent habitats and were unresponsive to aphid density in soybean. 

## 5. Conclusions

Our study shows coccinellids and *E. americanus* syrphids are likely the main groups contributing to soybean aphid suppression and is the first one to tease apart the factors affecting aphidophagous predator movement from adjacent habitats into soybean. Annual crops (canola, wheat) tended to have and provide more predators than perennial habitats (alfalfa, woody vegetation), probably because they harbored more resources for aphidophagous predators at the time of the study. Perennial crops and habitats are likely more important for predator populations early in the season before aphid establishment in annual crops. In general, our findings suggest that planting wheat and canola adjacent to soybean likely allows for the timely arrival of aphidophagous predators when aphid colonization is occurring. Future studies should investigate the functional role of these and other crops and habitats supporting predator populations and their movement into soybean at various times during the field season.

## Figures and Tables

**Figure 1 insects-14-00624-f001:**
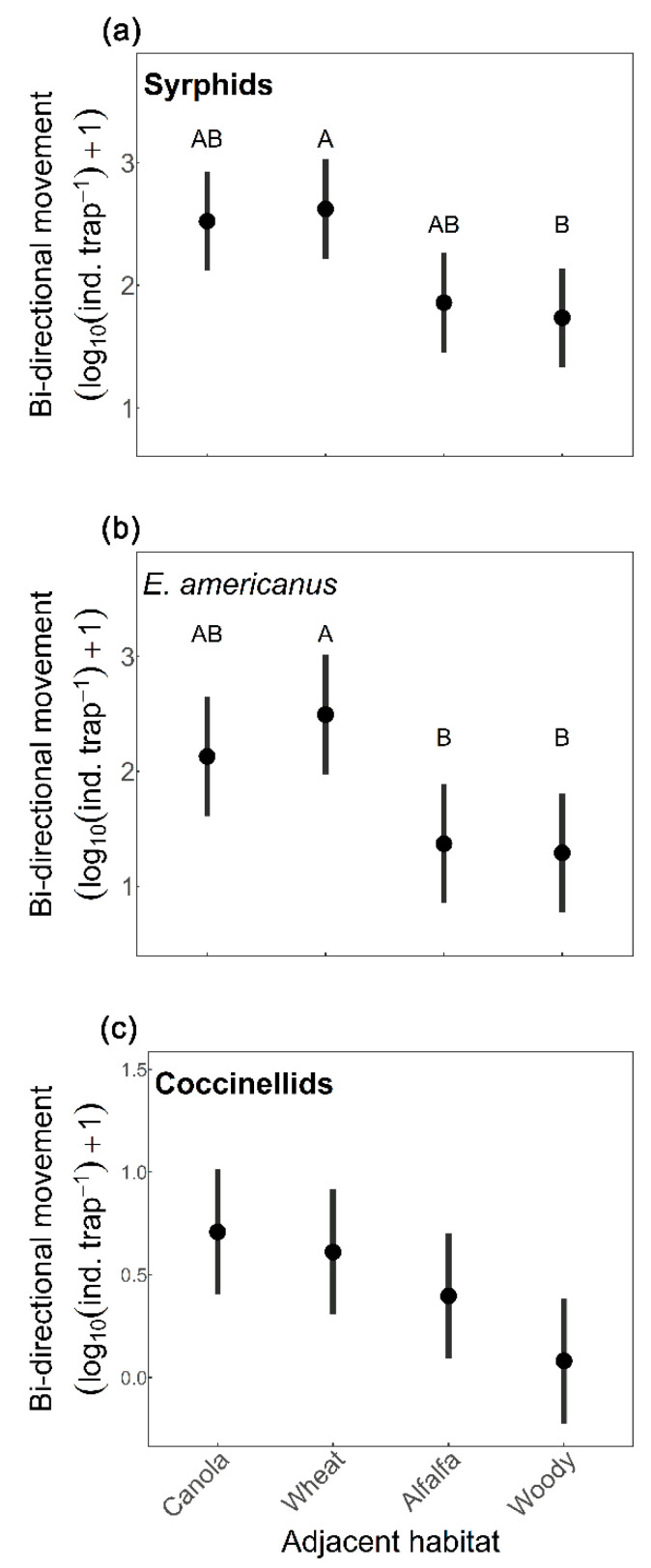
Effects of adjacent habitat type (canola, spring wheat, alfalfa, and woody vegetation) on predator bidirectional movement (log10 [individuals/trap] +1) in soybean, across two weeks collected from bidirectional Malaise traps, for (**a**) syrphids, (**b**) *Eupeodes americanus*, and (**c**) coccinellids. Confidence intervals (95%) were plotted. Different letters denote significant differences between adjacent habitat types (Tukey test; *p* < 0.05), see Table 2 for model statistics.

**Figure 2 insects-14-00624-f002:**
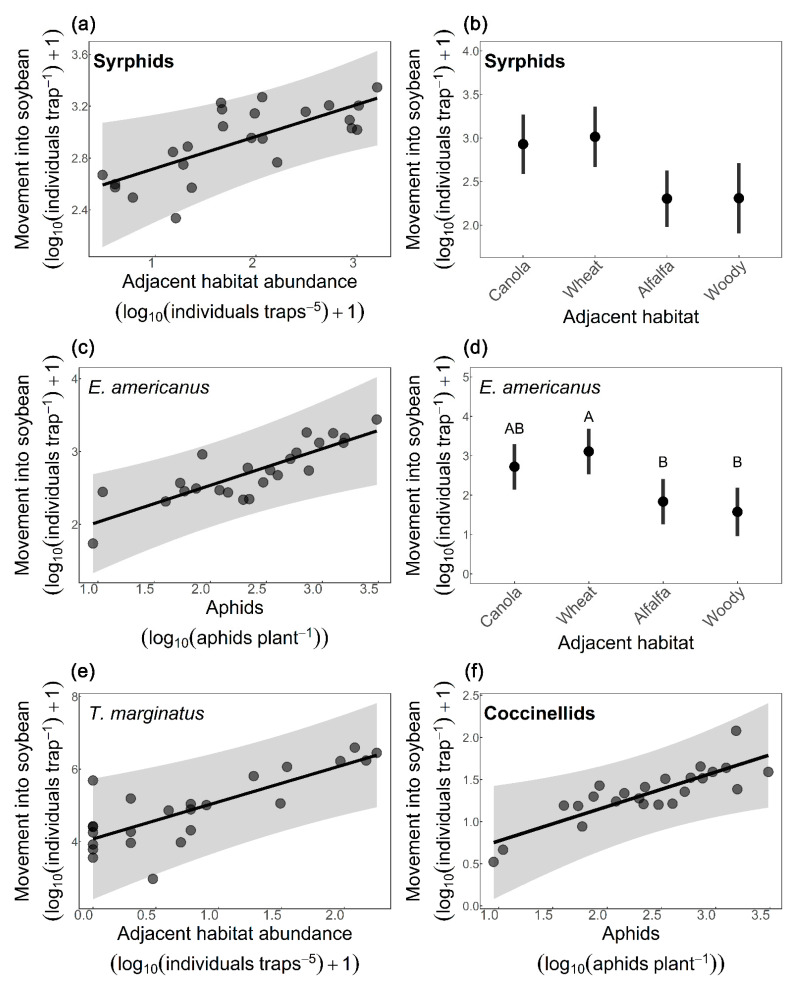
Factors affecting movement of predators into soybean (log10 [individuals/trap] +1) across two weeks collected from bidirectional Malaise traps. Effects of (**a**) number of syrphids in adjacent habitats and (**b**) adjacent habitat types on syrphid movement into soybean, of (**c**) number of aphids in soybean and (**d**) adjacent habitat types on *Eupeodes americanus* movement into soybean, of (**e**) number of *Toxomerus marginatus* in adjacent habitats on *T. marginatus* movement into soybean, and of (**f**) number of aphids in soybean on coccinellid movement into soybean. Adjacent habitat abundance was estimated using sticky traps (log10 [#/5 sticky traps] +1) and aphid abundance was estimated by plant counts (log10 [mean # aphids/plant]). Continuous variables in linear models are presented as black lines, partial residuals as gray dots and confidence intervals (95%) as grey area; categorical variables are presented as means ± standard errors. Different letters denote significant differences between adjacent habitat types (Tukey test; *p* < 0.05), see Table 3 for full model statistics.

**Table 1 insects-14-00624-t001:** Average predator abundance (± the standard error) in soybean and adjacent habitats across adjacent habitat types (alfalfa *n* = 3, canola *n* = 3, spring wheat *n* = 3, and woody vegetation *n* = 3) from sticky traps (#/5 sticky traps) collected the second sampling week. Statistics of linear models are presented in the text.

	Alfalfa	Canola	Wheat	Woody
Soybean				
Syrphids	529.7 ± 157.2	1809.0 ± 164.9	1085.7 ± 477.8	821.3 ± 575.8
*Toxomerus marginatus*	82.0 ± 40.3	287.3 ± 110.5	64.3 ± 22.4	201.3 ± 159.3
Coccinellids	36.0 ± 18.5	26.0 ± 5.2	50.7 ± 22.6	26.3 ± 20.9
All other predators	21.3 ± 8.7	30.7 ± 22.7	37.0 ± 18.7	21.7 ± 8.1
Adjacent habitat				
Syrphids	108.3 ± 6.2 a	1210.0 ± 188.8 b	525.7 ± 209.6 ab	302.0 ± 272.2 a
*Toxomerus marginatus*	18.3 ± 9.0 a	149.3 ± 17.3 b	43.0 ± 25.4 a	3.7 ± 1.3 a
Coccinellids	4.7 ± 2.4 ab	2.3 ± 0.9 ab	21.0 ± 4.4 b	3.3 ± 2.9 a
All other predators ^a^	7.7 ± 3.8 a	11.0 ± 4.4 a	15.0 ± 9.5 a	0.7 ± 0.7 a

^a^ All other predators includes anthocorids, chrysopids, hemerobiids, nabids, and staphylinids. Different letters within a row denote differences among adjacent habitats (Tukey test; *p* < 0.05).

**Table 2 insects-14-00624-t002:** The effect of bidirectional movement (movement into or out of soybean; df = 1, 34), week (1 or 2; df = 1, 34), and adjacent habitat (alfalfa, canola, spring wheat, woody vegetation; df = 3, 8) on predator bidirectional movement (log10 [individuals/direction/trap] +1) in soybean from linear mixed effects models. Estimates of fixed effects parameters (Est.) are presented.

	Direction			Week			Adjacent Habitat	
	Est.	F	*p*	Est.	F	*p*	F	*p*
Syrphids	0.19	4.28	0.046	0.44	22.0	0.0001	5.7	0.022
*Eupeodes americanus*	0.32	9.6	0.0039	0.34	11.1	0.0021	5.7	0.022
*Toxomerus marginatus*	0.024	0.043	0.84	0.85	52.7	0.0001		
Coccinellids	0.21	7.45	0.010	0.35	20.5	0.0001	3.9	0.055
All other predators ^a^	0.060	0.45	0.51	0.14	2.4	0.13		

^a^ All other predators includes anthocorids, chrysopids, hemerobiids, nabids, and staphylinids.

**Table 3 insects-14-00624-t003:** The effect of the field population of aphids in soybean (log10 [mean # aphids/plant]; df = 1, 10), predator abundance in the adjacent habitat (log10 [#/5 sticky traps] + 1); df = 1, 10 except *E. americanus* df = 1, 11), and adjacent habitat type (alfalfa, canola, spring wheat, or woody vegetation; df = 3, 8) on predator movement into soybean from linear mixed effects models. Adjacent habitat abundance refers to the response group’s abundance. Estimates of fixed effects parameters (Est.) are presented.

	Aphid Abundance			Adjacent Habitat Abundance			Adjacent Habitat	
	Est.	F	*p*	Est.	F	*p*	F	*p*
Syrphids	0.66	3.8	0.079	0.57	7.8	0.019	4.3	0.044
*Eupeodes americanus*	1.16	10.2	0.0085	-	-	-	6.5	0.015
*Toxomerus marginatus*	0.83	0.62	0.45	1.0	10.2	0.010	0.68	0.59
Coccinellids	0.94	6.6	0.028	0.47	1.8	0.21	3.7	0.062
All other predators ^a^	0.57	3.9	0.076	0.34	0.65	0.44	2.0	0.19

^a^ All other predators includes anthocorids, chrysopids, hemerobiids, nabids, and staphylinids.

## Data Availability

The data presented in this study are available on request from the corresponding author.

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
