# Peer review of "Annual Crops Contribute More Predators than Perennial Habitats during an Aphid Outbreak"

_insects, 2023, doi:10.3390/insects14070624_

Round 1

Reviewer 1 Report

The reviewed manuscript describes an interesting study on the movement of predators between habitats in relation to aphid population outbreak in one of those habitats (soybean crop). It would be reasonable to check whether the type of adjacent habitat affects not only the movement of predators, but also the subsequent aphid suppression. I assume, however, that this could be the subject of further research by the authors.

The manuscript is well written and I can only suggest a few minor edits:

(1) While mentioning Eupeodes americanus and Toxomerus marginatus in the abstract, it should be specified that they belong to Syrphidae, e.g. ‘Coccinellids and the syrphid Eupeodes …’ (l. 27).

(2) Individual graphs of figs 1 and 2 should be marked with the corresponding letters, especially as they are quoted in this way in the text (as e.g. fig. 1a, fig. 2b, etc.). Figure captions should also refer to those individually-named graphs.

(3) lines 345, 347, 348 and 350: change Coccinella tredecimpunctata to Hippodamia tredecimpunctata.

Author Response

PLEASE SEE OUR ANSWERS TO COMMENTS IN CAPITAL LETTERS AFTER EACH COMMENT. THANKS!

The reviewed manuscript describes an interesting study on the movement of predators between habitats in relation to aphid population outbreak in one of those habitats (soybean crop). It would be reasonable to check whether the type of adjacent habitat affects not only the movement of predators, but also the subsequent aphid suppression. I assume, however, that this could be the subject of further research by the authors.

The manuscript is well written and I can only suggest a few minor edits:

  • While mentioning Eupeodes americanus and Toxomerus marginatus in the abstract, it should be specified that they belong to Syrphidae, e.g. ‘Coccinellids and the syrphid Eupeodes …’ (l. 27).

CHANGE MADE

  • Individual graphs of figs 1 and 2 should be marked with the corresponding letters, especially as they are quoted in this way in the text (as e.g. fig. 1a, fig. 2b, etc.). Figure captions should also refer to those individually-named graphs.

LETTERS WERE ADDED TO THE FIGURES

  • lines 345, 347, 348 and 350: change Coccinella tredecimpunctata to Hippodamia tredecimpunctata.

CHANGES MADE: lines 346, 348, 349, 351. Thank you very much for spotting these mistakes!

Reviewer 2 Report

I do not often receive a high-quality well-written manuscript to review for Insects. I am glad to say that this manuscript describes a detailed and very well analyzed study on predator movement during an aphid outbreak in soybean. The study will be of considerable interest to readers of Insects.

I spotted only a few typos and an omission:

L21. Glycine (not plural)

L51 Suggest "...late phase of the growing season"

L61. Change have to has.

L68. Glycine

L130. ...in an area OF approximately...

Table 1. I assume that abundance values shown for soybean relate to bidirectional movement between soybean and each of the adjacent types of habitat. Please mention why these have not been compared statistically in the Title or a footnote.

L404. The word "driven" could be replaced with an alternative as causation was not demonstrated.

L431. coccinellids (all lowercase).

Some minor formatting issues in the References.

The Supplemental material is clear and supports the findings described in the main text.

Author Response

PLEASE SEE OUR RESPONSES IN CAPITAL LETTERS AFTER EACH COMMENT. THANKS!

I do not often receive a high-quality well-written manuscript to review for Insects. I am glad to say that this manuscript describes a detailed and very well analyzed study on predator movement during an aphid outbreak in soybean. The study will be of considerable interest to readers of Insects.

I spotted only a few typos and an omission:

L21. Glycine (not plural) CHANGE MADE, THANKS!

L51 Suggest "...late phase of the growing season" CHANGE MADE, THANKS!

L61. Change have to has. CHANGE MADE, THANKS!

L68. Glycine CHANGE MADE, THANKS!

L130. ...in an area OF approximately... CHANGE MADE, THANKS!

Table 1. I assume that abundance values shown for soybean relate to bidirectional movement between soybean and each of the adjacent types of habitat. Please mention why these have not been compared statistically in the Title or a footnote.

TABLE 1 PRESENTS THE RESULTS OF STICKY TRAPS AND THE STATISTICAL ANALYSES ARE PRESENTED IN THE TEXT, ALONG WITH LETTERS TO INDICATE SIGNIFICANT DIFFERENCES BETWEEN HABITATS IN THE TABLE. WE MODIFIED THE TITLE FOR MORE CLARITY:

Table 1. Average predator abundance (± the standard error) in soybean and adjacent habitats across adjacent habitat types (alfalfa n=3, canola n=3, spring wheat n=3, and woody vegetation n=3) from sticky traps (#/5 sticky traps) collected the second sampling week. Statistics of linear models are presented in the text. Different letters within a row denote differences among adjacent habitats (Tukey test; p<0.05).

L404. The word "driven" could be replaced with an alternative as causation was not demonstrated. DRIVEN WAS REPLACED BY “ASSOCIATED TO”:

L406: “…instead their movement was associated to the number of T. marginatus in the adjacent habitat.”

L431. coccinellids (all lowercase). CHANGE MADE, THANKS!

REFERENCES WERE FORMATTED CHANGING ABBREVIATIONS FOR JOURNAL NAMES, THANKS!

The Supplemental material is clear and supports the findings described in the main text.